# Neuroprotective and Anti—Neuroinflammatory Effects of a Poisonous Plant *Croton tiglium* Linn. Extract

**DOI:** 10.3390/toxins12040261

**Published:** 2020-04-17

**Authors:** Deepak Prasad Gupta, Sung Hee Park, Hyun-Jeong Yang, Kyoungho Suk, Gyun Jee Song

**Affiliations:** 1Department of Medical Science, College of Medicine, Catholic Kwandong University, Gangneung, Gangwon-do 25601, Korea; 2Department of Pharmacology, Brain Science and Engineering Institute, BK21 Plus KNU Biomedical Convergence Program, School of Medicine, Kyungpook National University, Daegu 41944, Korea; 3Department of Integrative Biosciences, University of Brain Education, Cheonan 31228, Korea; 4Translational Brain Research Center, International St. Mary’s Hospital, Catholic Kwandong University, Incheon 22711, Korea

**Keywords:** poisonous plant, neuroinflammation, *Croton tiglium* Linn., neuroprotection, M2 phenotype, microglia

## Abstract

Neuroinflammation is involved in various neurological diseases. Activated microglia secrete many pro-inflammatory factors and induce neuronal cell death. Thus, the inhibition of excessive proinflammatory activity of microglia leads to a therapeutic effect that alleviates the progression of neuronal degeneration. In this study, we investigated the effect of *Croton tiglium (C. tiglium)* Linn. extract (CTE) on the production of pro- and anti-inflammatory mediators in microglia and astrocytes via RT-PCR, Western blot, and nitric oxide assay. Neurotoxicity was measured by cell viability assay and GFP image analysis. Phagocytosis of microglia was measured using fluorescent zymosan particles. CTE significantly inhibited the production of neurotoxic inflammatory factors, including nitric oxide and tumor necrosis factor-α. In addition, CTE increased the production of the neurotrophic factor, brain-derived neurotrophic factor, and the M2 phenotype of microglia. The culture medium retained after CTE treatment increased the survival of neurons, thereby indicating the neuroprotective effect of CTE. Our findings indicated that CTE inhibited pro-inflammatory response and increased the neuroprotective ability of microglia. In conclusion, although CTE is known to be a poisonous plant and listed on the FDA poisonous plant database, it can be used as a medicine if the amount is properly controlled. Our results suggested the potential benefits of CTE as a therapeutic agent for different neurodegenerative disorders involving neuroinflammation.

## 1. Introduction

Neuroinflammation is observed in many neurological disorders, including Alzheimer disease (AD), stroke, multiple sclerosis, Parkinson’s disease (PD), and neuroinfections [1,2,3]. As innate immune cells in the central nervous system, microglia play a key role in regulating the pathogenesis of neurological disorders. Inflammatory activation of microglia (called as proinflammatory M1 microglia) increases neuroinflammation by releasing proinflammatory factors, including nitric oxide (NO), prostaglandin E2, tumor necrosis factor (TNF)-α, and interleukin (IL)-1β. These molecules are known to promote the progression of neurodegenerative diseases [4,5]. On the other hand, alternatively activated microglia (called anti-inflammatory M2 microglia) have neuroprotective properties that release neurotrophic factors (nerve growth factor or brain-derived neurotrophic factor; BDNF) and eliminate abnormal protein aggregation and pathogens [6,7,8]. Therefore, efforts are underway to identify natural materials and their target molecules that inhibit M1 inflammatory activation and promote M2 activation, and thus can be used as therapeutic agents for neurological diseases.

*Croton tiglium* (*C. tiglium*) Linn. extract (CTE) is the extract of the seed of this plant, which is an Asian herbal medicine. As a traditional medicine, *C. tiglium* is prescribed for many applications such as constipation, gastrointestinal disorders, intestinal inflammation, rheumatism, headache, and visceral pain. However, it is toxic at high doses [9,10,11,12]. Recent studies have investigated the antinociceptive effect, both in vivo and in vitro [13]. In these studies, the pain relief effect exerted by CTE was evaluated using the writhing test in mice, and six compounds were identified using high-performance liquid chromatography (HPLC). Moreover, CTE has been reported to exert antimicrobial and antidermatophytic properties [14,15]. Therefore, the ethanolic CTE has been used as a topical application, shampoo, or soap [14]. More recently, the antioxidant effect of CTE has been evaluated, and the efficiency of the extract was found to be enhanced after the incorporation of nanoparticles [16]. Antioxidant, pain relief, and anti-inflammatory properties are important features required in the treatment and prevention of many neurological diseases related to neuroinflammation. However, the anti-neuroinflammatory and neuroprotective properties of CTE have not yet been studied. Although CTE is known to be a poisonous plant and listed on the Food and Drug Administration (FDA) poisonous plant database, it can be used as a medicine if the amount is properly controlled.

In this study, we studied a novel function of CTE in the microglia. CTE was found to exert an anti-neuroinflammatory effect via the phenotypic switch toward the M2 anti-inflammatory and neuroprotective phenotype of microglia.

## 2. Results

### 2.1. Anti-Neuroinflammatory Effect of CTE

Microglia are the resident immune cells of the brain and are implicated in the regulation of synaptic pruning and neuronal networking in resting condition. However, under neuroinflammatory condition, activated microglia play a key role in the pathophysiology of many neurodegenerative diseases by releasing inflammatory and neurotoxic factors such as TNF-α, NO, and reactive oxygen species [17]. To identify potent anti-inflammatory and neuroprotective agents from natural materials, we searched the database of Korea Natural Herb Information and identified candidate materials based on lowering NF-κB activity from the NIKON bank (NIKOM Co. published in homepage). Among them, CTE showed strong anti-inflammatory effect in microglia. A microglia cell line, BV-2, was plated and stimulated with LPS (100 ng/mL) in the absence or presence of CTE (10 µg/mL). CTE significantly inhibited LPS-induced NO production but has no effect on microglial cell viability (Figure 1a,b). To confirm this anti-inflammatory effect, another microglia cells, namely, the HAPI cell line were used. Similarly, CTE significantly inhibited LPS-induced NO production, indicating the anti-inflammatory effect on HAPI cells. Furthermore, the half-maximal inhibitory concentration (IC50) of CTE was determined by constructing a dose–response curve and examining the effect of different concentrations of CTE on LPS-induced NO production (Figure 1c,d). 

### 2.2. Verification of the Anti-Neuroinflammatory Effect of CTE by Using Primary Microglia and Astrocytes

Next, we confirmed the anti-inflammatory effect of CTE in primary MGCs, because microglia and astrocytes are key players in neuroinflammation. The cells were cultured on 96-well plates and NO release was measured after 48 h of LPS treatment. CTE significantly inhibited LPS-induced NO release (Figure 2a) but has no effect on primary astrocytes and microglial cell viability. The expression of inflammatory cytokines was also studied. LPS increased the mRNA levels of *TNF-α* and *iNOS*, compared to the vehicle-treated group (Figure 2b). LPS-induced increase in proinflammatory cytokines was significantly suppressed by CTE treatment (Figure 2c,d).

### 2.3. Alternative Activation of Microglia

The inflammatory activation of microglia exerts neurotoxic effects, while alternative activation of microglia, called M2 type microglia, exerts neuroprotective functions such as phagocytosis and growth factor release [18]. To further investigate whether CTE could mediate alternative activation of microglia, we examined M2 activation. Primary mixed glial cells (MGCs) treated with CTE for 24 h, increased the expression of the M2 marker, *Arg1*, mRNA (Figure 3a,b), and protein (Figure 3c,d) compared to the vehicle group. CTE increased the expression of *Arg1* mRNA and protein, compared to the vehicle group. Similarly, the mRNA level of *BDNF*, a neurotrophic factor was significantly increased by CTE treatment in the microglia (Figure 3e). In addition, M2 polarization of microglia can be evaluated by phagocytic activity. Fluorescently labeled zymosan particles were significantly higher in CTE-treated microglia, compared to the vehicle group (Figure 3f). Our observation suggests that microglial polarization after CTE treatment leads to M2 activation.

### 2.4. Cytotoxicity Assay in Neuron

Since CTE is known to be a poisonous plant, we further examined cytoxicity of CTE in brain cell lines. Neuronal cells were treated with CTE in a dose-dependent manner, and cell viability was examined using the neuron cell lines N2a and B35 cells. CTE, however, has no effect on neuronal cell viability after either 24 h or 48 h of incubation with 5, 10, and 20 µg/mL CTE (Figure 4). 

### 2.5. Neuroprotective Effect of CTE

To confirm the neuroprotective effect of CTE, we next examined whether CTE could attenuate microglia-mediated neurotoxicity by using the neuroblastoma cells B35 (Figure 5a). Conditioned medium from LPS-treated microglia led to significant neuronal death. However, conditioned medium from CTE-treated microglia significantly decreased neuronal death (Figure 5b). Cell viability was measured by methylthiazolyldiphenyl-tetrazolium bromide (MTT) assay (Figure 5c) as well as by measuring the fluorescence intensity of EGFP (Figure 5d). Our results have demonstrated the treatment with CTE significantly increased neuronal survival suggesting CTE significantly prevented neuronal cell death from cytotoxic molecules produced from LPS-treated microglia. 

### 2.6. Anti-Neuroinflammatory Effect of CTE is Crotonic Acid-Independent

One of the major known chemicals in CTE is crotonic acid. Therefore, we examined the anti-neuroinflammatory effect of crotonic acid. HAPI cells were treated with crotonic acid and LPS. Crotonic acid did not inhibit LPS-induced NO release (Figure 6), it has no effect on microglial cell viability. Our result showed the anti-inflammatory effect of CTE is not derived from the single compound crotonic acid. 

### 2.7. Anti-Neuroinflammatory Effect of CTE Is MAPK-Dependent

To further examine the effects of CET on microglial activation pathway, we used the strategy of blocking specific signaling molecules. Inhibition of MAPK with PD98059, NF-κB with Bay 11-7082, ULK with SBI-0206965 significantly reduced LPS-induced increases in nitric oxide production suggesting all of these signaling molecules are important for LPS-induced inflammatory response in microglia. Interestingly, MAPK inhibitor, PD98059 abrogated the anti-inflammatory effect of CTE in microglia. While the anti-inflammatory effect of CTE was still observed in cells pretreated with Bay and SBI in primary mixed glial cells (Figure 7). These results indicate that CTE inhibits the LPS-induced production of inflammatory mediators mainly through the inactivation of MAPK pathways.

## 3. Discussion

Neurotoxic inflammatory factors released from activated microglia and astrocytes negatively affect brain health and cause neurodegenerative diseases, including AD and PD disease [19]. Microglia is a major cell type involved in neuroinflammation, and neuroinflammatory reactions are associated with neuronal death. Inflammatory activation of microglia (type M1) promotes the production of neurotoxins such as NO, TNF-α, and IL-1β, leading to neuronal cell death. On the other hand, alternative activation of microglia (type M2) promotes the nerve regeneration system and reduces neuroinflammation [20]. This study investigated the effect of CTE on neuroinflammation. First, the anti-neuroinflammatory effects of CTE were determined using the microglial cell lines BV-2 and HAPI. CTE significantly inhibited the production of neurotoxic inflammatory factors, including NO and TNF-α. In addition, CTE increased the expression of the pro-healing molecule Arg1 and neurotrophic factor BDNF. The culture medium retained from the CTE-treated microglia increased neuronal survival, thereby showing the neuroprotective effects of CTE.

*C. tiglium* has been used since long to treat different diseases such as diabetes, gastritis, and digestive disorders. *C. tiglium* has various biological and chemical implications such as anti-tumor, tumor-enhancing, anti-HIV, anti-inflammatory, anti-dermatophytic, and antioxidant activities [21]. In case of CNS diseases, CTE has been shown to reduce headache, seizures, and pain [13]. However, the direct effect of CTE on neurodegenerative diseases or neuroinflammation has not yet been investigated. Crotonoside, one of the key compounds in CTE, has been reported as a candidate for the treatment of acute myeloid leukemia (AML) [22]. In a previous study, crotonoside was found to inhibit AML cell proliferation by inhibiting HDAC3/6/NF-κB pathway. The HDAC inhibitor valproic acid is known to attenuate traumatic spinal cord injury-induced inflammation via the STAT1 and NFκB pathways [23]. A limitation of our study is that the molecular mechanism of the anti-inflammatory effect of CTE was not examined. However, it can be suggested that the anti-inflammatory effects of CTE are caused, at least in part, by the inhibition of HDAC/NF-κB pathway [22,23] and MAPK activity (Figure 7).

*C. tiglium* seed contains large proportion of croton oils. Crotonic acid is one of the major compounds of CTE, however crotonic acid is not a key compound for the anti-neuroinflammatory effect of CET (Figure 6). *C. tiglium* seed contains large proportion of croton oils. Croton oil is the source of phorbol derivatives, in particular 12-O-tetradecanoylphorbol-13-acetate (TPA), which is known as a potent cell proliferation enhancer. Thus, it is expected that the CTE may act like TPA, which is known to exert neuroprotective properties such as neurite outgrowth and growth cone formation [24,25]. Recently, anti-tumor and antioxidant compounds have been isolated from *C. tiglium* seeds and characterized by spectroscopic analysis [26]. CTE is also known to induce the production of vascular endothelial growth factor (VEGF). VEGF is a secreted mitogen associated with angiogenesis [27]. VEGF was long thought to be a potent neurotrophic factor for neuronal survival. These studies, therefore, suggest a neuroprotective effect of CTE. In one study, pro-inflammatory effect of *C. tiglium* L. have been reported [28]. Pro-inflammatory cytokines, including *TNF-α* and *IL-1β*, were significantly increased in the macrophage cell line RAW264.7 treated with crude proteins of *Croton tiglium* L., and the inflammatory effects was associated with p38-MAPK. In their study, crude protein extracted from *C. tiglium* L. with petroleum ether was used for the inflammatory reaction, which is different from our CTE extracted with the methanol-based method showing anti-inflammatory effects.

Microglia, the local immune cell type of the CNS, are widely distributed in the brain and spinal cord. Their roles are recognized as the prime components of the CNS inflammatory system. Astrocytes are the most abundant glia in the brain, and astrocytic activation also contributes to and expands the initial inflammatory response and neurodestructive effects of microglia [29,30]. Although initial studies on neuroinflammation suggested that microglial activation was detrimental to neurogenesis [31,32,33], more recent studies have proven the beneficial role of microglial activation on the regulation of the migration, proliferation, and differentiation of neural stem cells [34,35]. In particular, alternative activation of microglia (type M2) can increase neuroprotective factors such as nuclear factor erythroid 2-related factor 2 and heme oxygenase-1 [36]. In addition, M2 microglia can produce the anti-inflammatory cytokine IL-10 and inhibit neuroinflammation [37,38]. It was also shown that BDNF-secreting microglial cells found in the subventricular zone (SVZ) of the adult mouse brain are highly proliferative and facilitate nerve regeneration by promoting the migration of neuroblasts to the site of injury [39]. Therefore, the discovery of new drugs promoting microglial polarization toward the M2 phenotype has become a topic of interest for the development of potential therapeutic and preventive strategies for neurodegenerative diseases [2,18,40].

## 4. Conclusions

In summary, CTE significantly inhibited the production of neurotoxic inflammatory factors, including NO and TNF-α. CTE increased the production of the neurotrophic factor, BDNF and phagocytosis in microglia. In addition, CTE increased the survival of neurons protected from neurotoxic factors released from LPS-stimulated microglia, thereby indicating the neuroprotective effect of CTE. Although CTE is known to be a poisonous plant, it can be used as a medicine if the amount and the purification method are properly controlled. The anti-neuroinflammatory and neuroprotective effects of CTE might be beneficial for neurodegenerative diseases such as AD and PD.

## 5. Materials and Methods

Cell culture media and antibiotics, such as HyClone Dulbecco’s Modified Eagle Medium (DMEM), fetal bovine serum (FBS), penicillin, and streptomycin, were purchased from GE Healthcare Life. Lipopolysaccharide (LPS; Escherichia coli, L2880), methylthiazolyldiphenyl- tetrazolium bromide (MTT; M2128), and poly-L-lysine (p6407), crotonic acid (113018) were purchased from Sigma Aldrich (Saint Louis, Missouri, USA). pHrodo^®^ Red Zymosan Bioparticles (P35364) were purchased from Thermo Fisher Scientific (Carlsbad, CA, USA). Antibodies against arginase 1 (NB100-59730; Novus Biologicals, Briarwood Avenue Centennial, CO, USA) and β-actin (MA5-15739; Thermo Fisher Scientific, Rockford, IL, USA) were also purchased.

CTE preparation. CTE was obtained from the National Development Institute of Korean Medicine; it exerts an inhibitory effect on NF-κB activation (Homepage, Natural Herb Material Bank, Daegu, Korea). Briefly, the seed of *C. tiglium* was refined, dried, and ground into fine powder. The powder was refluxed twice with 3.5 L of 70% MeOH (sample/MeOH ratio, 1:7; w/w). The extract was filtered through filter paper and concentrated. The extracts were concentrated under reduced pressure, and an extract was obtained through a freeze-drying process. The extract was dissolved in DMSO at 100 mg/ml and used as a stock solution.

Cell culture. BV-2 cells, an immortalized mouse microglial cell line [40], were maintained in DMEM containing 5% FBS and 50 μg/mL gentamicin at 37 °C. A highly aggressively proliferating immortalized (HAPI) rat microglial cell line [41], B35 neuroblastoma cell line, N2a mouse neuroblastoma cell line [42], and mouse primary microglia and astrocytes were maintained in DMEM containing 10% FBS, 10 U/mL penicillin, and 10 μg/mL streptomycin at 37 °C in a humidified 5% CO2 incubator. Animal-based and experimental procedures were approved by the Institutional Review Board of the Kyungpook National University School of Medicine and performed according to the guidelines of the NIH Guide for the Care and Use of Laboratory Animals. The primary microglial and astrocyte mixed cultures were prepared from mouse brain, as previously described with minor modifications [43]. In brief, the brains of 3-day-old C57BL/6 mice were chopped and dissociated by mechanical disruption by using a cell strainer (size, 70-µm pores). The cells were plated onto poly-L-lysine-coated plates.

Measurement of nitric oxide (NO). NO production in cell culture media was estimated by measuring the amount of nitrite, a stable NO metabolite. Cells were plated on 96-well plates (40,000 cells/well) and treated for 24 h with 100 ng/mL LPS in the presence or absence of CTE. At the end of a 24-h incubation period, the cell culture medium (50 μL) was mixed with an equal volume of Griess reagent (5% phosphoric acid, 0.1% naphthylethylenediamine dihydrochloride and 1% sulfanilamide) in a 96-well plate. Absorbance was read at 540 nm by using a microplate reader. Standard curves were prepared based on the reference values of serially diluted sodium nitrite solution.

Assessment of cell viability. Cell viability was determined using modified MTT assay. The culture medium was aspirated after LPS treatment for 24 h, with or without the compounds. MTT (0.5 mg/mL in phosphate-buffered saline; PBS) was added to the cells, and the mixture was incubated at 37 °C for 2 h. The resulting formazan crystals were dissolved in DMSO (Dimethyl sulfoxide). Absorbance was determined at 570 nm.

Traditional and real-time polymerase chain reaction (RT-PCR). Total RNA was extracted from the treated cells by using TRIZOL reagent (Invitrogen, Carlsbad, CA, USA), according to the manufacturer’s protocol. cDNA is synthesized from total RNA using the Superscript II reverse transcriptase (Invitrogen) and an oligo (dT) primer. Traditional polymerase chain reaction (PCR) was performed with specific primer sets as shown in Table 1 using a T100 Thermal Cycler (Bio-Rad, Richmond, CA, USA). After PCR, 10 µL of each PCR product was electrophoresed on a 2% agarose gel. DNA fragments were detected under ultraviolet light following ethidium bromide staining. As the internal reference gene, glyceraldehyde 3-phosphate dehydrogenase (*GAPDH)* was used. Quantitative real-time PCR was performed using the One Step SYBR PrimeScript RT-PCR Kit (Takara Bio, Otsu, Shiga, Japan), according to the manufacturer’s instructions, followed by detection using the ABI Prism 7000 Sequence Detection System (Applied Biosystems, California, CA, USA).

Western blot analysis. The cells were washed with cold PBS after various treatments and lysed with RIPA lysis buffer (50 mM Tris-HCl, 150 mM NaCl, 0.02% sodium azide, 0.1% SDS and 1% NP-40). Equal amounts of protein were separated on 10% SDS-polyacrylamide gel and transferred to polyvinylidene difluoride (PVDF) membranes (Bio-Rad Laboratories, California, USA). The blots were blocked with 4% skim milk in PBS with 0.1% Tween-20 (PBST), and then incubated with primary antibodies (goat anti-Arginase1, 1:500 dilution; Novus, Biologicals, Inc., Littleton, CO, USA) and mouse anti-β-actin (1:5000 dilution; Sigma-Aldrich, Saint-Louis, MO, USA) overnight at 4 °C. After thorough washing with PBST, horseradish peroxidase-conjugated secondary antibodies (1:2000) in 5% skim milk were applied for 1 h at room temperature. The blots were developed using an enhanced chemiluminescence detection kit (SuperSignal™ West Femto; Thermo Fisher, Franklin, MA, USA).

Neurotoxicity assay. Green fluorescent protein (GFP)-transfected B35 rat neuroblastoma cells (B35-GFP) were used to study the neurotoxicity of CTE, as previously described [43]. In brief, rat microglia cells, namely, HAPI cells, (4 × 10^4^/well in 96-well plate) were treated with LPS (100 ng/mL) in the presence or absence of 10 μg/mL CTE for 6 h and washed with PBS. Then, the cells were cultured in DMEM for 48 h. The culture medium was then stored and used as conditional medium for neurotoxicity assay. B35-GFP neuroblastoma cells (1 × 10^4^/well) were incubated with conditional medium for 48 h. At the end of incubation, the number of viable B35-GFP cells in five randomly chosen fields per well was counted, and GFP images were captured by fluorescence microscopy. Neuronal cell survival was measured by MTT assay. N2a, that is, mouse neuroblastoma cells, were used to confirm the absence of neurotoxicity in the cells treated with different concentrations of CTE.

Phagocytosis of fluorescent zymosan particles. Primary mixed glial cells (MGC) were seeded at a density of 2 × 10^4^ cells/well in 96-well plates and cultured for 48 h. Cells were treated with LPS (1 μg/mL) and/or CTE (10 μg/mL) for 1 h in serum-free DMEM. Fluorescent zymosan particles (30 μg/mL; zymosan A from Saccharomyces cerevisiae) were added into cells and incubated at 37 °C for 3 h (BioParticles conjugated with Photo; Molecular Probes Inc., Carlsbad, CA, USA). Cells were then washed three times with ice-cold PBS to remove unbound particles. Photomicrographs of five randomly chosen fields were captured under a fluorescent microscope (Nikon ECLIPSE 80i, Nikon Corp, Nishioi, Shinagawa-ku, Tokyo, Japan).

Statistical analysis. Data represented as the mean ± standard error of mean (SEM) from three or more independent experiments unless stated otherwise. *p*-values of < 0.05 were considered statistically significant. To check the statistical significance of three or more groups, one-way ANOVA was used to compare the values, followed by Tukey’s multiple comparison test. For the comparison of two groups, unpaired two-tailed Student’s t-test was used using GraphPad Prism software (version 8). All data of this work are available from the corresponding authors on request.

## Figures and Tables

**Figure 1 toxins-12-00261-f001:**
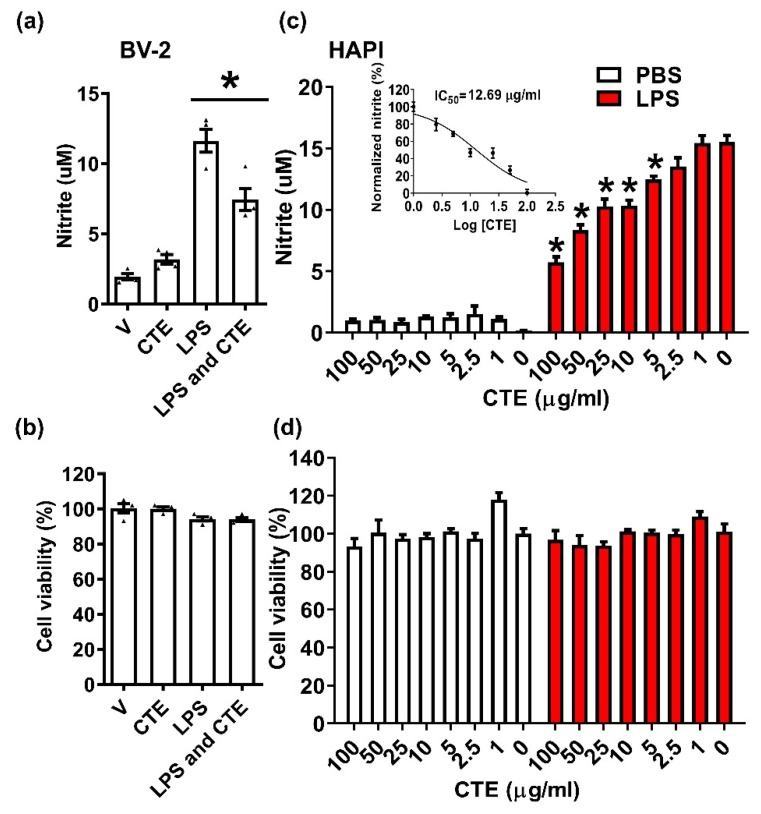
Anti-inflammatory effect of CTE in the microglia. The anti-inflammatory effect of CTE was measured in the mouse microglial cell line BV-2 cells. (**a**) BV-2 cells were stimulated with lipopolysaccharide (LPS, 100 ng/mL), in the presence or absence of CTE (10 μg/mL). The level of nitric oxide (NO) in the culture medium was measured 24 h post treatment. LPS-induced NO production was significantly reduced by CTE treatment. (**b**) Cytotoxicity was measured by methylthiazolyldiphenyl-tetrazolium bromide (MTT) assay 24 h after treatment with LPS, in the presence or absence of CTE. (**c**) Dose-response curve and IC50 value for CTE-treated microglia. Dose-dependent effect of CTE was observed in the microglial cell line HAPI. Graph for IC50 calculation, with the x-axis indicating CTE (log transformation), and y-axis, normalized nitrite concentration (percent maximum value). IC50 was calculated by non-linear regression analysis. (**d**) Cell viability after stimulation with LPS and CTE in HAPI cells. MTT was used to measure cell viability. N = 3 independent experiments/group. Each black filled Δ represents an individual value. Values represent the mean ± SEM, * *p* < 0.05 vs. LPS-only, from ANOVA.

**Figure 2 toxins-12-00261-f002:**
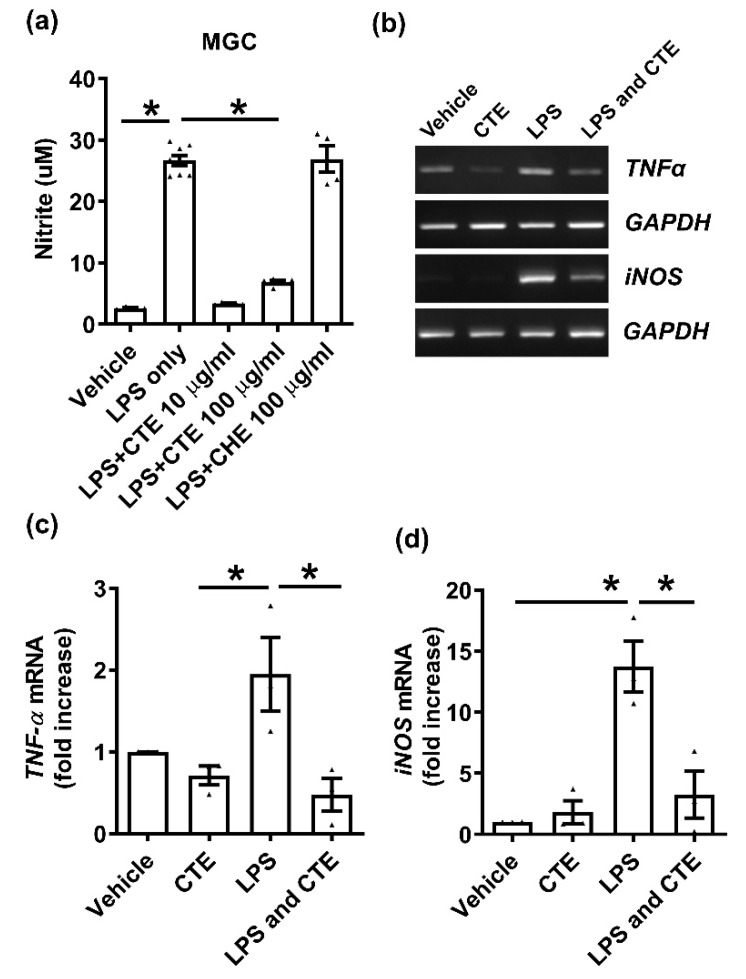
Functional validation of CTE as an anti-inflammatory material in mouse mixed glial cells (MGC). (**a**) The MGC were stimulated with lipopolysaccharide (LPS) (1 μg/mL) and/or CTE. Nitric oxide (NO) production induced by LPS (1 μg/mL for 48 h) were significantly reduced by CTE treatment (10 or 100 μg/mL). Control herb extract (CHE, 100 μg/mL) was used as a negative control. The Control Herb Extract was obtained from *Cucucrbita moschata*. * *p* < 0.05 compared to LPS only. (**b**) mRNA expression of the microglial activation (M1) markers, *TNF-α*, and *iNOS* was measured using RT-PCR. MGC was treated with LPS (1 μg/mL) in the presence or absence of CTE (10 μg/mL) for 48 h. *GAPDH* was used as a loading control. (**c**,**d**) *TNF-α* and *iNOS* mRNA expression was quantified using ImageJ, based on three independent experiments. Each black filled Δ represents an individual value. Values represent the mean ± SEM, * *p* < 0.05 from ANOVA.

**Figure 3 toxins-12-00261-f003:**
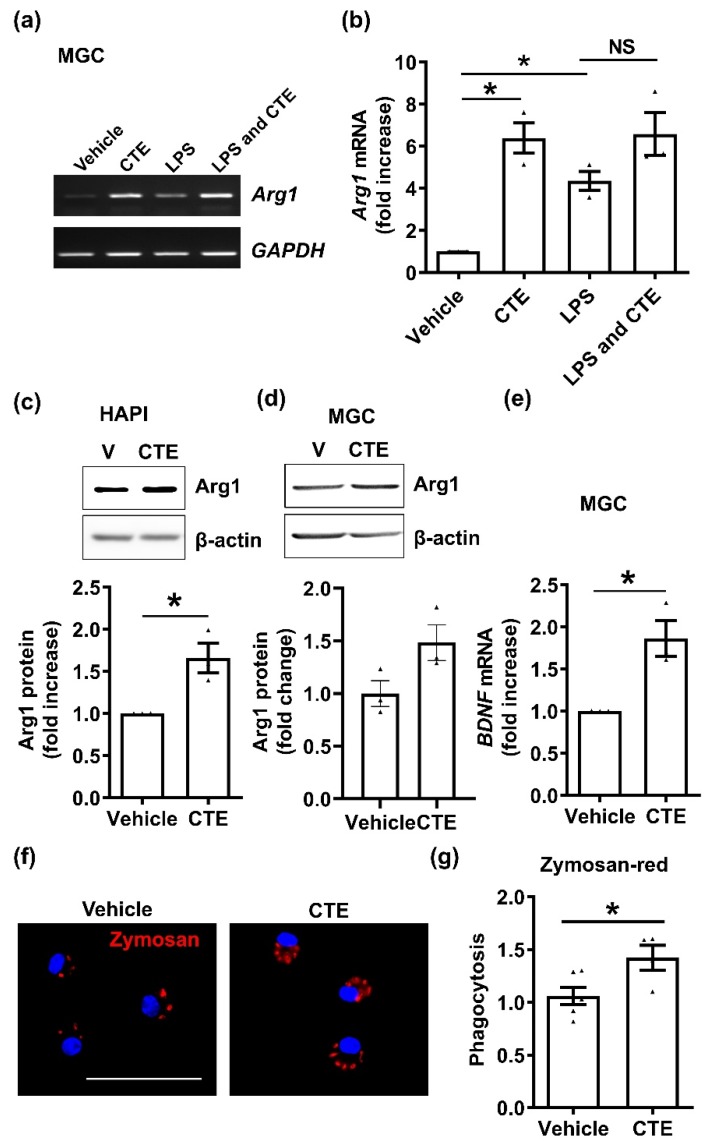
CTE promotes M2 polarization of microglia in vitro. (**a**) mRNA expression level of the M2 marker *Arg1* in primary mixed glial cells (MGC). Total RNA was extracted from 10 µg/mL CTE-treated MGC or/and LPS (1 µg/mL) for 48 h. *Arg1* mRNA expression was measured using RT-PCR. *GAPDH* was used as a loading control. (**b**) *Arg1* mRNA expression was quantified using ImageJ, based on three independent experiments. (**c**) Expression of Arg1 protein in the microglial cell line HAPI. Cells were treated with CTE (10 µg/mL) for 24 h, and cell lysates were used for western blotting. Beta-actin was used as loading control. (**d**) Expression of Arg1 protein in MGC. Cells were treated with CTE (10 µg/mL) for 24 h, and cell lysates were used for western blotting. Beta-actin was used as loading control. The band intensity was measured by ImageJ and normalized by loading control (**e**) *BDNF* mRNA expression levels in MGC. Cells were treated with CTE for 48 h, and mRNA levels were quantified using real-time PCR. (**f**) Representative image for zymosan-red. Scale bar 50 µm. (**g**) Zymosan-red phagocytosis assay in MGC. Cells were plated and treated with CTE (10 µg/mL) or vehicle (0.1% DMSO) for 48 h. Opsonized zymosan-fluorescent particles were added into MGC and further incubated for 3 h. After three rounds of washing, phagocytosed zymosan particles were measured by fluorescence microscopy. Fold increase of phagocytosis per vehicle is shown graphically. N ≥ 3 independent experiments/group. Each black filled Δ represents an individual value. Values represent the mean ± SEM, * *p* < 0.05 was considered significant, by using Student’s t-test or one-way analysis of variance (ANOVA). NS, not significant

**Figure 4 toxins-12-00261-f004:**
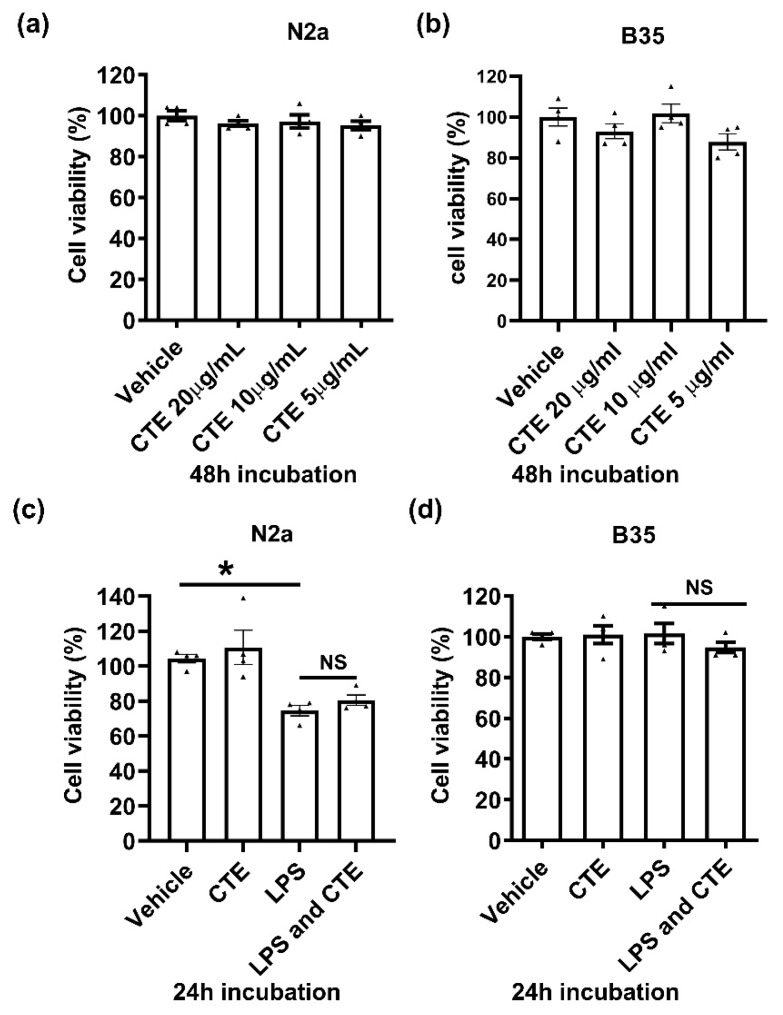
Cytotoxicity assay in neuronal cells treated with CTE. Cytotoxicity of CTE was analyzed using the neuroblastoma cell lines N2a and B35 cells). Cells were treated with different concentrations of CTE (**a**,**b**) and/or LPS (100 ng/ml) (**c**,**d**) as indicated and incubated for indicated hours. Cell viability was measured by MTT assay. N = 3~4 independent experiments/group. Each black filled Δ represents an individual value. Value represent the mean ± SEM. * *p* < 0.05 was considered significant, by using Student’s t-test or one-way analysis of variance (ANOVA). NS, not significant.

**Figure 5 toxins-12-00261-f005:**
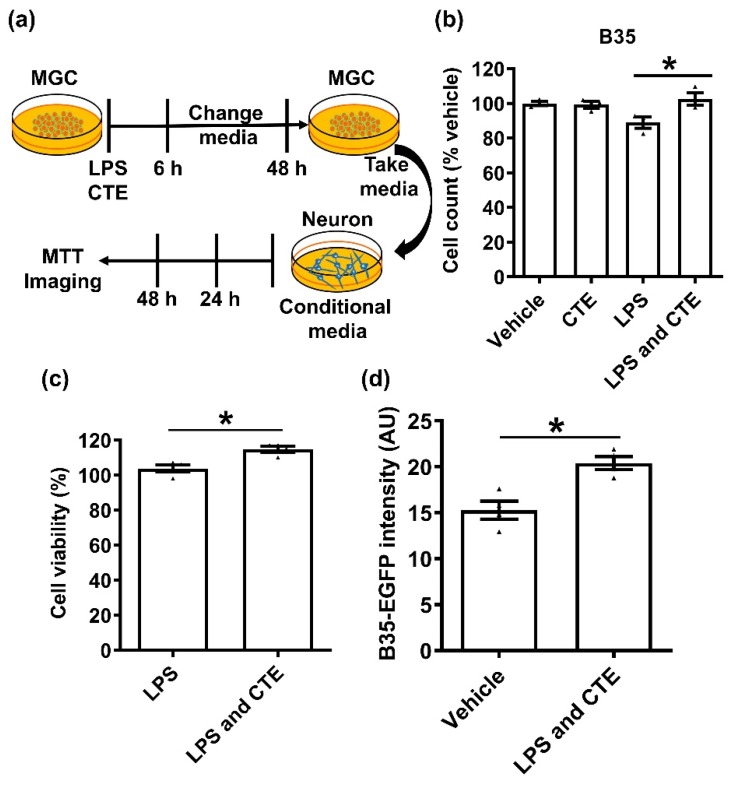
Neuroprotective effects of CTE on microglia-mediated neurotoxicity. (**a**) Schematic diagram of the experimental timeline. LPS (1 μg/mL), CTE (10 μg/mL), and conditional media for MGC and neurons. MGC were treated with vehicle, LPS (1 µg/mL), CTE (10 µg/mL), or LPS/CTE and incubated for 6 h. Then, the medium was changed and incubated for 48 h. The medium, called as conditional medium, was retained for neuron toxicity assay. The neuroblastoma cell line B35 cells stably expressing green fluorescent protein (B35-EGFP) were cultured in conditional medium. After further incubation for 48 h, cell count determination (**b**), MTT assay (**c**), and fluorescent microscopy (**d**) were performed for viability assessment. N = 3 independent experiments/group. Each black filled Δ represents an individual value. Value represent the mean ± SEM. * *p* < 0.05 was considered significant, by using Student’s t-test or one-way analysis of variance (ANOVA).

**Figure 6 toxins-12-00261-f006:**
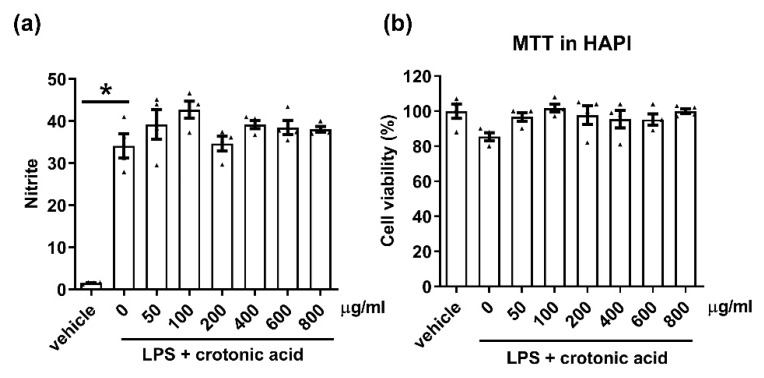
No anti-inflammatory effect of crotonic acid in the microglia. The anti-inflammatory effect of crotonic acid was measured in the microglial cell line HAPI cells. (**a**) Cells were stimulated with lipopolysaccharide (LPS, 100 ng/mL), in the presence or absence of crotonic acid (50~800 μg/mL). The level of nitric oxide (NO) in the culture medium was measured 24 h post treatment. LPS-induced NO production was measured by Griess solution. (**b**) Cytotoxicity was measured by MTT assay 24 h after treatment with LPS, in the presence or absence of crotonic acid. N = 3 independent experiments/group. Each black filled Δ represents an individual value. Values represent the mean ± SEM. * *p* < 0.05 was considered significant, by using Student’s t-test or one-way analysis of variance (ANOVA).

**Figure 7 toxins-12-00261-f007:**
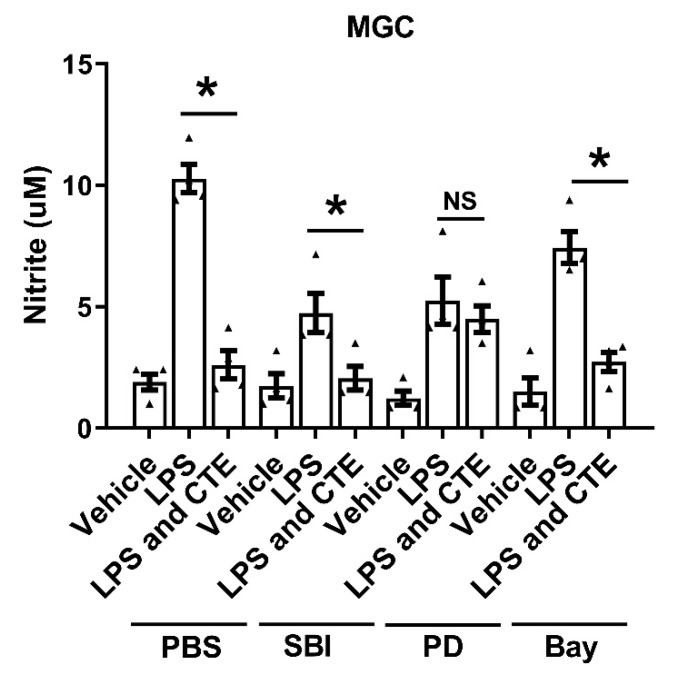
MAPK activation is required for the anti-inflammatory effect of CTE. The anti-inflammatory effect of CTE was measured in primary mixed glial cells (MGC). MGC were pretreated with inhibitors (PD 10 μM, SBI 5 μM, and Bay 2.5 μM) for 1 h and stimulated with lipopolysaccharide (LPS, 1 μg/mL) and CTE (10 μg/ml) for 48 h. The level of nitric oxide (NO) in the culture medium was measured 48 h post treatment. N = 3 independent experiments/group. Each black filled Δ represents an individual value. Values represent the mean ± SEM, * *p* < 0.05 vs. LPS-only, from ANOVA.

**Table 1 toxins-12-00261-t001:** Primers used for RT-PCR.

Target Genes	Accession Number	Forward Primer (5’-3’)	Reverse Primer (5’-3’)	Temp (°C)	Cycles
*TNF-α*	NM_013693.2	CATCTTCTCAAAATTCGAGTGACAA	ACTTGGGCAGATTGACCTCAG	60	25
*iNOS*	NM_010927.3	CCCTTCCGAAGTTTCTGGCAGCAGC	GGCTGTCAGAGCCTCGTGGCTTTGG	70	30
*Arg-1*	NM_007482	CGCCTTTCTCAAAAGGACAG	CCAGCTCTTCATTGGCTTTC	60	29
*BDNF*	NM_007540.4	CGCAAACATGTCTATGAGGGTTC	TAGTAAGGGCCCGAACATACGAT	60	30
*GAPDH*	NM_008084	ACCACAGTCCATGCCATCAC	TCCACCACCCTGTTGCTGTA	60	25

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
