# Peer review of "Neuroprotective and Anti—Neuroinflammatory Effects of a Poisonous Plant Croton tiglium Linn. Extract"

_toxins, 2020, doi:10.3390/toxins12040261_

Round 1

Reviewer 1 Report

In this manuscript, the authors describe a number of in vitro experiments to study the neuroprotective and anti-neuroinflammatory effects of an extract from the plant Croton tiglium. Overall, I found the manuscript to be well done. The scientific soundness of the experiments is good with proper controls for all studies. I also found the writing to be well done and easy to read and understand. I recommend publishing the manuscript after a few minor changes. It would be interesting to see how well the CTE could provide neuroprotection in an animal model. Hopefully that is in future plans.

In this study the authors indicate and demonstrate in their models that the CTE has anti inflammatory properties. However, there are reports in the literature that Croton tiglium has pro inflammatory properties. There is no mention of that in the manuscript. This is an oversight, that needs to be addressed. The authors need to state the confounding studies, cite the papers, and then attempt to explain the reasons why they have observed different results in this study. I feel that this is a major issue that has to be addressed.

The last sentence (lines 251-252) in the conclusion paragraph needs to be removed.

In the material and methods section, line 263, please explain further what is meant by the term “refined”. Additionally, in line 265, please explain with more detail how the samples were concentrated.

Author Response

Response to Reviewer 1 Comments

In this study the authors indicate and demonstrate in their models that the CTE has anti-inflammatory properties. However, there are reports in the literature that Croton tiglium has pro inflammatory properties. There is no mention of that in the manuscript. This is an oversight, that needs to be addressed. The authors need to state the confounding studies, cite the papers, and then attempt to explain the reasons why they have observed different results in this study. I feel that this is a major issue that has to be addressed. The last sentence (lines 251-252) in the conclusion paragraph needs to be removed.

Response: Thank you very much for your comments. We have started in vivo experiment in two different diseases models. As you know it takes time to complete. We hope that our next manuscript will provide the neuroprotective effect of CTE in these animal models. We also thank you for pointing out the pro-inflammatory effect of Croton tiglium L. in previously published article. As reviewer’s comment, we mentioned the previously published report on the proinflammatory effect of crude CTE protein (in Discussion, line 236~241) and added this reference (28). Honestly, the results of the paper (reference 28) are not reliable, so we intentionally tried to add a very brief explanation of their findings. For example, the graph (fig 7) and the description in text is not consistent.  In their paper, [the effects of crude protein on the MAPK signaling pathway were examined. p38‑MAPK, p‑p38‑MAPK, ERK1/2, p‑ERK1/2, JNK1‑3 and p‑JNK1‑3 protein levels were decreased in RAW264.7 cells upon croton proteins treatment (0, 25, 50 and 100 μg/ml) (Fig. 7).] However, Fig 7 showed the significant increase of p-38/total-p38 and no changes in p-ERK/total ERK and p-JNK.

The last sentence (lines 251-252) in the conclusion paragraph needs to be removed.

The last sentence (lines 251-252) in the conclusion paragraph has been removed.

In the material and methods section, line 263, please explain further what is meant by the term “refined”. Additionally, in line 265, please explain with more detail how the samples were concentrated.

The term used “refined” in the manuscript means “removal of the impurities from the seeds of Croton tiglium Linn. The detail for the concentration is addressed in line 284~286. [The extracts were concentrated under reduced pressure, and an extract was obtained through a freeze-drying process. The extract was dissolved in DMSO at 100 mg/ml and used as a stock solution.]

Revised version was uploaded.

Reviewer 2 Report

In the paper entitled: “Neuroprotective and anti-neuroinflammatory effects of a poisonous plant Croton tiglium Linn. Extract” authors evaluated anti-neuroinflammatory effect of Croton tiglium extract (CTE) on microglia and neuronal cells. Results obtained are very interesting and, despite the limitations due to the lack of in vivo experiments, these preliminary data are very useful for future studies. The introduction is well structured, concise and provides all the necessary background. The research design is appropriate and experimental procedures have been well conducted.
However, there are some points that need to be clarified.

Chapter 2.1: In line 65-66 authors state that CTE was used in BV-2 experiments at both 10 and 100 μg/ml but only 10 μg/ml dose is shown in Figure 1a.
Why did not authors perform a dose-response curve in BV-2 cells likewise in HAPI cells? This would have made it possible to calculate an IC50 for both cell lines.
In cell line experiments an end point of 24 hours was chosen while primary MGC experiments an end point of 48 hours was chosen. Anyway, LPS dose used in cell line experiments was 100 ng/ml while in primary MGC experiments was 1 μg/ml. Please, explain.

Chapter 2.2: What is the composition of the control herb extract (CHE) used as a control in the experiment shown in figure 2a? Please specify there or in Material section.

Chapter 2.3: authors evaluated the neuroprotective effects of CTE on microglia. Why have all the experiments been carried out on MGCs except for the Western blot which was performed on HAPI cells? Figure 3f should also show the image of the vehicle.

Chapter 2.4: In Figure 4, panel C, cited in the legend to the figure, is lacking. The concentration of CTE used in B35 cells (panel 4b) is not reported and it is not clear why the experimental conditions are different in N2a cells with respect to B35 (for example, LPS treatment is missing in N2a cells, three different concentrations tested in N2a cells and only one concentration in B35 cells).
I think that the last sentence of the legend to Figure 4 is a misprint.

Chapter 2.5: author state that conditioned medium from CTE-treated microglia significantly decreased neuronal death in B35 cells, but in Figure 5b, statistical significance is not reported.

Chapter 2.7: Experiments reported in panel 7b are not commented and it is not clear their meaning.

Conclusions need to be rewritten, better underlining the remarkable aspects obtained in the work (the last sentence is a misprint).

In Material and Methods section, information about neuroblastoma N2a cell line are lacking.
In western blot analysis, the dilution of secondary antibody is missing

Round 2

Reviewer 1 Report

good.

Reviewer 2 Report

Authors have fully satisfied all my comments. In my opinion, the work is now better focused and comprehensive.

I propose publication on Toxins in the present form